# Clinical Aspects and Significance of β-Chemokines, γ-Chemokines, and δ-Chemokines in Molecular Cancer Processes in Acute Myeloid Leukemia (AML) and Myelodysplastic Neoplasms (MDS)

**DOI:** 10.3390/cancers16193246

**Published:** 2024-09-24

**Authors:** Jan Korbecki, Mateusz Bosiacki, Piotr Stasiak, Emilian Snarski, Agnieszka Brodowska, Dariusz Chlubek, Irena Baranowska-Bosiacka

**Affiliations:** 1Department of Anatomy and Histology, Collegium Medicum, University of Zielona Góra, Zyty 28, 65-046 Zielona Góra, Poland; jan.korbecki@onet.eu (J.K.); p.stasiak@inz.uz.zgora.pl (P.S.); 2Department of Biochemistry and Medical Chemistry, Pomeranian Medical University in Szczecin, Powstańców Wlkp. 72, 70-111 Szczecin, Poland; mateusz.bosiacki@pum.edu.pl (M.B.);; 3Institute of Medical Sciences, Collegium Medicum, University of Zielona Góra, Zyty 28 Str., 65-046 Zielona Góra, Poland; emiliansnarski@gmail.com; 4Department of Gynecology, Endocrinology and Gynecological Oncology, Pomeranian Medical University in Szczecin, Unii Lubelskiej 1, 71-252 Szczecin, Poland; agnieszka.brodowska@pum.edu.pl

**Keywords:** leukemia, acute myeloid leukemia (AML), chemokine, CCL22, CCL25, bone marrow, myelodysplastic neoplasms (MDS)

## Abstract

**Simple Summary:**

This article examines the significance of β-chemokines, γ-chemokines, and δ-chemokines in acute myeloid leukemia (AML). It focuses on the effects of these chemotactic cytokines on both leukemic cells and non-leukemic cells within the tumor niche in the bone marrow. This article emphasizes the substantial impact of certain chemokines on tumorigenic processes in AML, highlighting the correlation between chemokine expression and patient prognosis. However, the mechanisms underlying this relationship remain poorly understood. The lack of comprehensive understanding of the role of chemokines in AML impedes the development of new anti-leukemic drugs targeting these chemokines and their receptors.

**Abstract:**

Background/Objectives: Acute myeloid leukemia (AML) is a type of leukemia with a very poor prognosis. Consequently, this neoplasm is extensively researched to discover new therapeutic strategies. One area of investigation is the study of intracellular communication and the impact of the bone marrow microenvironment on AML cells, with chemokines being a key focus. The roles of β-chemokines, γ-chemokines, and δ-chemokines in AML processes have not yet been sufficiently characterized. Methods: This publication summarizes all available knowledge about these chemotactic cytokines in AML and myelodysplastic neoplasm (MDS) processes and presents potential therapeutic strategies to combat the disease. The significance of β-chemokines, γ-chemokines, and δ-chemokines is detailed, including CCL2 (MCP-1), CCL3 (MIP-1α), CCL5 (RANTES), CCL23, CCL28, and CX3CL1 (fractalkine). Additionally, the importance of atypical chemokine receptors in AML is discussed, specifically ACKR1, ACKR2, ACKR4, and CCRL2. Results/Conclusions: The focus is on the effects of these chemokines on AML cells, particularly their influence on proliferation and resistance to anti-leukemic drugs. Intercellular interactions with non-AML cells, such as mesenchymal stem cells (MSC), macrophages, and regulatory T cells (T_reg_), are also characterized. The clinical aspects of chemokines are thoroughly explained, including their effect on overall survival and the relationship between their blood levels and AML characteristics.

## 1. Introduction

Acute myeloid leukemia (AML) is a type of leukemia derived from hematopoietic stem cells [1,2]. Consequently, the highest accumulation of AML cells is found in the bone marrow [3], from where they are released into the bloodstream. In 14% of patients, AML cells inhabit other organs; in such cases, they are referred to as extramedullary AML [4]. The most commonly affected tissues include the skin, central nervous system (CNS), and pleura. Rarely, AML can also affect other organs, such as lymph nodes, pericardium, liver, spleen, and testes.

AML is classified based on genetic abnormalities [2]. It is classified into AML with *PML:RARA*, *RUNX1:RUNX1T1*, *CBFB:MYH11*, *DEK:NUP214*, *RBM15:MRTFA*, *BCR:ABL1* fusion, or those with *KMT2A*, *MECOM*, or *NUP98* rearrangement, as well as with mutations in the *NPM1* or *CEBPA* genes. There is also AML with other defined genetic alterations and AML with myelodysplasia-related changes (*AML-MRC*). AML with other defined genetic alterations includes rare genetic abnormalities or those recently discovered, whereas AML-MRC involves genetic abnormalities associated with myelodysplasia.

AML-MRC is closely related to myelodysplastic neoplasms (MDS), which is a precursor stage of this subtype of AML. MDS is characterized by blood cytopenias and bone marrow dysplasia [5] and typically occurs in older adults, with a prevalence of 45 cases per 100,000 older individuals. MDS often progresses to AML-MR. The boundary between MDS and AML-MRC is fluid, but the 20% threshold of bone marrow blasts is generally recognized as the defining point [2].

In particular, AML from this group features a complex karyotype with at least three abnormalities, including del(5q), del(11q), del(12p), del(17p), and i(17q), along with somatic mutations in genes such as *ASXL1*, *BCOR*, *EZH2*, *SF3B1*, *SRSF2*, *STAG2*, *U2AF1*, and *ZRSR2* [2].

Other genetic abnormalities in AML cells may be significant in the treatment of patients, for example mutations in the FMS-like tyrosine kinase-3 (*FLT3*) gene [6]. These mutations increase the tyrosine kinase activity of FLT3, stimulating AML cell proliferation. Approximately 30% of AML patients exhibit mutations in this gene [7]. Therefore, FLT3 inhibitors are being tested in AML patients with *FLT3* gene mutations [8].

In cases where no genetic abnormalities are classified, AML is categorized based on the differentiation of AML cells [2]. There are eight AML types defined by this differentiation, based on specific differentiation markers assessed either within or on AML cells. For example, antigens such as CD13, CD33, and CD117 are associated with AML with minimal differentiation, AML without maturation, and AML with maturation. Monocytic markers of acute monocytic leukemia include antigens CD11c, CD14, CD36, and CD64, along with non-specific esterase activity.

Classification of AML by differentiation began in the late 1970s when the French–American–British (FAB) cooperative group introduced a system based on morphological traits [9,10,11]. Examples of this classification include AML without maturation (AML-M1) and monocytic AML (AML-M5). Since the original FAB classification did not cover rarer types of AML, AML-M0 (AML with minimal differentiation) and AML-M7 (acute megakaryoblastic leukemia) were later added [10,11].

Today, classification based on differentiation is gradually being phased out [2], replaced by systems based on genetic abnormalities. This newer approach categorizes AML according to genetic changes driving the cancerous behavior of AML cells. It has the potential to link specific targeted therapies aimed at genetic mutations with corresponding AML subtypes, guiding more precise treatment strategies.

The highest incidence of AML occurs in North America, Western Europe, and Australia, with approximately 2.5 out of 100,000 people diagnosed with AML annually in these regions [12,13]. The disease is seldom diagnosed in East Asia. For instance, in Nanjing, China, it is estimated that 1.35 in every 100,000 citizens are diagnosed with AML [14]. The highest incidence of AML is noted in the elderly, with the median age of patients exceeding 60 years [12,15]. Men are more frequently affected by AML than women [13].

The prognosis for AML patients is poor, with a median overall survival of only 8.5 months [15]. Only 32% of AML patients survive two years post-diagnosis. Due to the very unfavorable prognosis, new therapeutic approaches to AML treatment are being explored, particularly the interaction of AML cells with the bone marrow microenvironment. One component of this microenvironment includes chemokines.

Chemokines are characterized as cytokines that exhibit chemotactic activity, primarily stimulating immune cell migration [16]. In inflammatory reactions, where chemokine production is particularly heightened, various immune cells infiltrate the affected area. In humans, 43 different chemokines can be distinguished and grouped into four sub-families according to the conserved cysteine motif at the N-terminus [16]:α-chemokines with a CXC motif at the N-terminus. In humans, 16 representatives can be distinguished: CXC chemokine ligand 1-17 (CXCL1-17),β-chemokines with a CC motif at the N-terminus. In humans, 24 representatives can be distinguished: CC chemokine ligand 1-28 (CCL1-28),γ-chemokines with an XC motif at the N-terminus. In humans, 2 representatives can be distinguished: XC chemokine ligand 1-2 (XCL1-2), andδ-chemokines with a CX3C motif at the N-terminus. In humans, 1 representative can be distinguished: CX3C chemokine ligand 1 (CX3CL1).

Chemokines act through their specific receptors, with each receptor activated by one or more chemokines from only one subfamily [16]. Thus, chemokine receptors are named according to the sub-family of chemokines their ligand belongs to. For example, CC chemokine receptor (CCR)1, CCR2, and CCR3 are receptors for β-chemokines, while CX3CL1 is a receptor for CX3C chemokine receptor 1 (CX3CR1).

Chemokine receptors are part of the G-protein-coupled receptor (GPCR) family [17]. A key signal transduction pathway from these receptors involves the activation of heterotrimeric G proteins. This activation triggers signal transmission to phospholipase C-β (PLC-β), phosphatidylinositol-4,5-bisphosphate 3-kinase (PI3K), and, in the case of Gαi, reduces protein kinase A (PKA) activity. PI3K activates Akt/PKB, which exerts anti-apoptotic effects. PLC-β raises cytoplasmic Ca^2+^ levels, driving cell migration. Chemokine receptor activation also triggers G protein-independent signaling pathways, such as JAK/STAT and pathways involving small GTPases. Activation of small GTPases promotes actin polymerization and cell migration. MAPK cascades are also activated, leading to increased cell proliferation (Figure 1) [17].

A notable exception is CX3CR1 [18]. The motif in CX3CR1, essential for G protein coupling, differs from that of other chemokine receptors. Consequently, signal transduction to G proteins is diminished compared to other receptors. CX3CR1 also functions as an adhesion protein, binding to the membrane-bound form of CX3CL1.

Chemokines are involved in the mechanisms of numerous diseases, including solid cancers [19], and they play a role in tumorigenesis in leukemias. The best-understood chemokine in AML is CXCL12 and its receptor, CXCR4. Currently, drugs targeting this chemokine receptor are being tested [20,21,22]. The roles of other chemokines in AML tumorigenesis have not been as thoroughly studied and are not a popular target of research.

AML cells with FAB M4–M5 phenotypes show elevated expression of several chemokines and chemokine receptors [23,24]. This may be linked to epigenetic changes (e.g., DNA methylation) and increased expression and activation of various transcription factors in these cells compared to other AML phenotypes. For instance, NF-κB [25], a key transcription factor in inflammatory responses, is notably active. Another example is PU.1/SPI1, which is most highly expressed in AML cells with the FAB M5 phenotype [23,24]. PU.1/SPI1 drives the expression of CCL22 in macrophages and dendritic cells (DC) [26], potentially explaining the elevated CCL22 levels in AML cells with a monocytic phenotype [23,24]. Nevertheless, the direct causes of increased chemokine and chemokine receptor expression in AML cells across different phenotypes are seldom explored experimentally.

This paper was written for two main reasons. First, there is currently no review available that discusses the importance of β-chemokines, γ-chemokines, and δ-chemokines in AML. This paper aims to compile all available information about these chemokines in AML tumorigenesis to assist researchers in planning experiments to explore their significance in AML. Second, this paper aims to highlight gaps in current knowledge and emphasize the importance of chemokines in AML tumor processes, which hinder the development of new anti-leukemic drugs.

In this work, the importance of the chemokines and chemokine receptors in question was investigated using public databases of gene expression profiling interactive analysis (GEPIA) (http://gepia.cancer-pku.cn) [27] and the University of Alabama at Birmingham cancer (UALCAN) (https://ualcan.path.uab.edu) [23,24]. The aim was to link the expression levels of these chemokines and receptors in AML cells to the prognosis of patients with this type of leukemia. In every demonstration of such an association, the need for further research on the respective chemokine or chemokine receptor was emphasized.

The numbering of chemokines is not ranked according to the numbering of their receptors. To prioritize all the knowledge collected, this article is divided into three parts, according to the sub-families discussed. When discussing β-chemokines, the chemokines were grouped by the activated receptor responsible for their most important properties and then ranked by receptor numbering.

## 2. Information Retrieval Method

Experimental articles for this review were retrieved using the PubMed search engine (https://pubmed.ncbi.nlm.nih.gov/, accessed 1 May 2024). This enabled a search across all articles in the MEDLINE database that included the target terms in the title, abstract, or keywords. The search was conducted without restrictions on publication date. The following search phrase was used:

(ccl1 or ccl2 or ccl3 or ccl4 or ccl5 or ccl6 or ccl7 or ccl8 or ccl9 or ccl10 or ccl11 or ccl12 or ccl13 or ccl14 or ccl15 or ccl16 or ccl17 or ccl18 or ccl19 or ccl20 or ccl21 or ccl22 or ccl23 or ccl24 or ccl25 or ccl26 or ccl27 or ccl28 or ccr1 or ccr2 or ccr3 or ccr4 or ccr5 or ccr6 or ccr7 or ccr8 or ccr9 or ccr10 or ccr11 or i-309 or mcp-1 or mip-1a or mip-1alfa or mip-1b or mip-1beta or rantes or mcp-2 or mcp-3 or eotaxin or mcp-4 or mcp-5 or hcc-1 or hcc-2 or tarc or parc or mip-3a or mip-3alfa or mip-3b or mip-3beta or elc or larc or exodus or slc or mdc or mpif-1 or mpif-2 or teck or mec or ctack or eskine or xcl1 or xcl2 or xcr1 or xcr2 or cx3cl1 or cx3cr1 or fractalkine or neurotactin or lymphotactin) aml leukemia not review.

Thus, the selected articles had to contain the following search terms:The articles were required to include the name of a chemokine. In the 1990s and the early 2000s, chemokines were often named by their discoverers. In 2000, a new classification system was established [28], where the name of each chemokine in a given subfamily included the motif (CC, CXC, CX3C, XC), followed by an indicator of whether it was a chemokine/ligand (L) or receptor (R), and a number designating the chemokine or receptor. To ensure comprehensive results, searches were conducted using chemokine names from both the old and new nomenclature across the relevant subfamilies.The articles had to include the abbreviation of the disease being studied, “AML”. Since the focus of the review is on the significance of specific chemokine subfamilies in AML, the selected articles needed to contain both the name of a chemokine and the abbreviation of the disease.The articles needed to include the term “leukemia”. In various fields, the abbreviation “AML” can refer to concepts other than “acute myeloid leukemia”, such as “renal angiomyolipoma”, “alveolar macrophage-like”, the AML-12 cell line (hepatocyte), or “adversarial machine learning”, among others. To filter out articles where “AML” was used in contexts unrelated to “acute myeloid leukemia”, the search focused on identifying those that contained both the abbreviation “AML” and the word “leukemia”.All review articles were excluded from the search. Although reviews often cite experimental studies, they typically present only a portion of the data from those sources. To gain a more comprehensive understanding of the topic, only experimental studies were considered for analysis. However, for writing brief introductions to specific chemokine axes, review articles were sometimes used and cited to provide readers with general background information on chemokines.Additionally, when searching for articles on chemokines in MDS, the search focused on articles that included the name of the target chemokines along with the terms “MDS” and “myelodysplastic”.

## 3. β-Chemokines

### 3.1. Ligands of the CCR1 Receptor

CCR1 (CD191) is a receptor for several β-chemokines, including CCL3, CCL4, CCL5, CCL7, CCL13, CCL14, CCL15, CCL16, and CCL23 [16]. Some of these ligands also activate other β-chemokine receptors, notably CCR3 and CCR5.

#### 3.1.1. CCR1 in AML

CCR1 expression is higher in AML cells compared to cord blood-derived CD34^+^ cells [29]. More detailed studies have shown that CCR1 expression is elevated in AML cells with the FAB M4–M5 phenotypes compared to the less differentiated FAB M0–M1 subtypes [23,24,30], and the expression levels are similar to those observed in monocytes [30]. In contrast, CCR1 expression levels in AML cells with the FAB M0-M2 phenotypes are comparable to those in CD34^+^ cells derived from bone marrow [30]. Additionally, CCR1 receptor expression is higher in CD34^+^ AML cells than in CD34^−^ AML cells [31]. Increased CCR1 expression in AML is associated with a poorer prognosis [32], indicating that this receptor and its ligands play a significant role in AML tumorigenesis.

#### 3.1.2. CCL3 in AML

CCL3 is a ligand for CCR1, CCR3, and CCR5 [16,33]. Previously known as macrophage inflammatory protein-1α (MIP-1α) and LD78α [28,34], it is a crucial pro-inflammatory factor, playing a significant role in immune system functioning, particularly as a chemoattractant for various lymphocytes [35]. CCL3 is also important in the progression of AML.

Adult AML patients have elevated blood levels of CCL3 compared to healthy individuals [36,37,38]. Chemotherapy treatment generally reduces this chemokine level to that observed in healthy subjects; however, one study indicates that AML patients may have lower blood levels of CCL3 than healthy individuals [39]. Additionally, elevated levels of CCL3 are observed in the bone marrow of adult AML patients, along with elevated levels of CCL3-like-1 (CCL3L1) [40], when compared to healthy individuals. Similarly, CCL3 expression in bone marrow cells is higher than in a control group [36]. In patients with MDS, serum levels of CCL3 are elevated compared to healthy individuals [41]. Similarly, CCR1 expression is higher in bone marrow CD34^+^ cells of MDS patients than in healthy controls [42]. This indicates a potential role and activation mechanism of this receptor in the oncogenic pathways associated with MDS.

AML cells secrete CCL3 (Figure 2) [30,31,36,43,44,45,46]. Cells with the FAB M4–M5 phenotypes exhibit higher CCL3 secretion than those with the less differentiated FAB M0–M1 phenotypes [23,24], although some research does not indicate this dependency [30]. AML cells with the FAB M4–M5 phenotypes can secrete more CCL3 than monocytes [30], while AML cells with the FAB M0–M1 phenotypes produce less CCL3 than bone marrow CD34^+^ cells [30]. One study shows that although CCL3 mRNA is expressed in AML cells, the chemokine is not observed at the protein level [47]. AML cells may exhibit a constitutive ability to produce this chemokine [44]. Myeloid ecotropic integration site-1 (MEIS1) can increase CCL3 expression in AML cells [48,49], and physiological hypoxia in the bone marrow may also elevate CCL3 production in AML cells [43,50,51]. Interactions between mesenchymal stem cells (MSCs) and AML cells can further increase CCL3 production in bone marrow [45,52]. Macrophages and granulocytes may also contribute to CCL3 levels in the bone marrow [46].

AML cells that secrete high amounts of CCL3 also secrete notable quantities of CCL2, CCL4, CXCL1, and CXCL8/interleukin-8 (IL-8) [31]. These AML cells exhibit higher activity of nuclear factor-κB (NF-κB), which is responsible for the increased production of these chemokines [31].

CCL3 is involved in various pathological mechanisms in AML, making it significant in the disease’s progression. Higher CCL3 expression in AML cells is associated with a poorer prognosis for patients [37]. CCL3 can affect AML cells directly by either inhibiting or increasing their proliferation, depending on the study and patient [31,53,54,55,56]. In most patients, CCL3 may not affect AML cell proliferation at all.

CCL3 is also active in the bone marrow, which is crucial in AML pathology. It induces regulatory T cell (T_reg_) recruitment and accumulation in bone marrow, likely through CCR1 and CCR5, as shown in in vivo experiments with mice [46]. CCL3 cooperates with CXCL12 in this process. Although the mechanism of T_reg_ recruitment in AML has not been thoroughly investigated in humans, AML patients show much higher expression of CXCR4 (the receptor for CXCL12) in T_reg_ compared to healthy individuals [57], suggesting that the CXCL12–CXCR4 axis may be more significant for T_reg_ recruitment in AML patients than CCL3. T_reg_ suppress the immune system’s response, which, under pathological conditions, can inhibit the immune system’s efforts to fight cancer cells, including AML [58]. Recruited T_reg_ in bone marrow suppress the immune response against AML, particularly inhibiting the action of CD8^+^ T cells [46]. For this reason, T_reg_ may reduce the effectiveness of the patient’s immune system in combating AML as well as the efficacy of immunotherapy [59]. Consequently, an increase in T_reg_ numbers in the bone marrow worsens the prognosis for AML patients [46,60].

In a few percent of AML cases, AML cells are found in other organs, a phenomenon known as extramedullary leukemia [61]. Chemokines can recruit AML cells to various organs from the blood. CCL3 likely contributes to the development of extramedullary AML of the skin, particularly in pediatric patients, through its interaction with CCR5 [62].

AML patients often show abnormal erythrocyte test results due to CCL3’s activity, which disrupts erythropoiesis [63,64]. Proper hematopoiesis requires osteoclasts, and CCL3 disrupts their function, impairing erythropoiesis [36]. CCL3 also acts directly on megakaryocyte-erythroid progenitors (MEPs), reducing their proliferation via the CCR1 receptor and activation of p38 mitogen-activated protein kinase (MAPK) [37]. This reduces the formation of megakaryocytes and platelets, explaining the reduced platelet count in AML patients [63,64]. CCL3 also induces apoptosis in erythroblasts, further disrupting erythropoiesis and reducing erythrocyte and platelet counts [37]. This behavior is similar to that of another CCR1 ligand, CCL23, suggesting the two chemokines complement each other in inhibiting erythropoiesis in AML patients [40]. This could also explain the higher levels of CCL3 in the serum of MDS patients, a condition marked by cytopenias.

#### 3.1.3. CCL23 in AML

CCL23 is a ligand for CCR1 [16,65] and shares similar properties with other ligands that act through the CCR1 receptor, such as CCL3 [37,40]. Previously known as myeloid progenitor inhibitor factor (MPIF)-1 and CKβ8 [28,34], CCL23’s increased expression in AML cells is associated with a tendency towards poorer prognosis (*p* = 0.059) according to data from UALCAN (https://ualcan.path.uab.edu accessed 21 June 2023) [23,24], suggesting clinical relevance for AML patients. CCL23 levels are elevated in the bone marrow of both adult [40] and pediatric AML patients [66] compared to healthy individuals.

AML cells may not be the primary source of this chemokine, as only approximately half of adult AML patients exhibit detectable, low production of CCL23 in AML cells [31]. The highest expression of CCL23 is found in AML cells with the FAB M4–M5 phenotypes [23,24]. The production of this chemokine may correlate with the production of other chemokines, including CCL5, CXCL9, CXCL10, and CXCL11 [31].

CCL23 has diverse effects on AML cell proliferation. In some adult patients, it increases proliferation, while in others, it decreases proliferation [31]. In most patients, CCL23 does not affect AML cell proliferation [31]. It also does not affect the proliferation of the U-937 cell line [67].

The most significant trait of CCL23 in AML patients is its effect on hematopoiesis. CCL23 inhibits hematopoiesis in the bone marrow by affecting colony-forming unit-granulocyte, macrophage (CFU-GM), CFU-GM size, and mixed colonies (CFU-GEMM) [40,68]. It may also affect burst-forming unit-erythroid (BFU-E) [68].

#### 3.1.4. Other CCR1 Ligands: CCL14, CCL15, and CCL16 in AML

Currently, there are no comprehensive studies on the significance of CCL14, CCL15, and CCL16 in AML. These chemokines likely play no significant role in the progression of AML. While in some cases, these chemokines may increase AML cell proliferation in adult patients, most patients with this cancer experience no effect on AML cell proliferation from these chemokines [31]. Additionally, studies of pediatric patients found no differences in blood CCL14 levels between AML patients and healthy subjects [69]. In adult patients, CCL14 expression is lower in AML cells with the *FLT3* gene mutation compared to AML cases without this mutation (Table 1) [23,24].

### 3.2. Ligands of the CCR2 Receptor

CCR2 (CD192) ligands include monocyte chemotactic proteins (MCP): CCL2 (MCP-1), CCL7 (MCP-3), CCL8 (MCP-2), and CCL13 (MCP-4). These chemokines can also activate other receptors, including CCR1, CCR3, and CCR5 [16,70]. As their name suggests, this chemokine group consists of monocyte chemoattractants responsible for monocyte recruitment and the accumulation of tumor-associated macrophages (TAM) in solid tumors [71]. The significance of MCP in AML is under investigation. However, the expression levels of CCR2, CCL2, CCL8, and CCL13 in AML cells are not associated with patient prognosis [23,24], suggesting that the involvement of CCR2 and its ligands in AML development may not be clinically relevant.

#### 3.2.1. Expression of CCR2 and Its Ligands in AML

There is higher expression of CCR2 in AML cells compared to cord blood-derived CD34^+^ cells [29]. However, other studies indicate that CCR2 expression in AML cells is lower than in monocytes [72]. CCR2 expression in AML cells varies depending on the leukemia phenotype, with the highest expression found in AML cells with the FAB M4–M5 phenotypes [23,24,30]. This expression is higher than in AML cells with the FAB M0–M2 phenotypes and comparable to monocyte expression [30]. This may be related to the myelomonocytic and monocytic phenotypes of FAB M4 and M5 AML cells, respectively [9]. CCR2 expression is also higher in AML cells with the inv(16) mutation compared to those without this mutation [29], and in CD34^+^ AML cells compared to CD34^−^ AML cells [31]. CCR2 expression varies across different cell lines. For example, THP-1 cells exhibit high CCR2 expression, while U-937 and Kasumi cell lines show low expression [72].

Adult AML patients also exhibit higher blood levels of CCL2 compared to healthy individuals [73]. AML patients with the FAB M4–M5 phenotypes tend to have the lowest CCL2 levels among all FAB subtypes [73], though some studies show no difference in CCL2 levels relative to the FAB classification [74]. Contrarily, one study found no difference or even lower CCL2 levels in AML patients compared to healthy individuals [39,72]. Higher levels of CCL2 have been observed in the bone marrow plasma of AML patients compared to healthy individuals [75], though another study found the opposite [72].

The CCL2 level in the blood of AML patients may depend on the AML type and serve as a predictive factor in therapy. Lower blood CCL2 levels are associated with fewer AML blasts in the blood [74]. Chemotherapy does not significantly affect elevated CCL2 levels in AML patients, but a higher post-chemotherapy CCL2 level than at diagnosis may indicate a high likelihood of relapse [74]. Lower CCL2 levels suggest disease remission post-chemotherapy [39], and higher CCL2 levels may predict graft vs. host disease (GVHD) following bone marrow transplantation and response to immunotherapy [74].

AML cells often produce high quantities of CCL2 [31,44], with production being higher in cells with the FAB M4–M5 phenotypes compared to the FAB M0–M1 phenotypes [30]. However, some studies do not confirm this dependency [23,24]. All AML cells with the FAB M4–M5 phenotypes produce CCL2, while only approximately half of the AML cells with the FAB M0–M1 phenotypes produce CCL2, at lower levels than bone marrow CD34^+^ cells [30].

The level of CCL2 production in AML cells may also be influenced by specific mutations. CCL2 expression is lower in AML cells with the *FLT3* gene mutation compared to those without this mutation [23,24]. Additionally, LncRNA LINC01255 reduces CCL2 expression in AML cells, a process dependent on B Lymphoma Mo-MLV Insertion Region 1 (BMI1) [76].

AML cells produce other MCP as well. Approximately half of AML patients exhibit very low CCL7 production in their AML cells [31]. CCL7 production is present in AML cells with the FAB M4–M5 phenotypes but absent in those with the FAB M0–M1 phenotypes [30]. CCL8 is produced by AML cells in one-third of patients with the FAB M4–M5 phenotype, similar to monocytes, but not by AML cells with the FAB M0–M1 phenotypes [30]. CCL8 production may be highest in AML cells with the FAB M6 phenotype [23,24].

In half of AML patients, AML cells produce very low amounts of CCL13 [31], while other studies show no CCL13 production [30]. The highest production of CCL8 may occur in AML cells with the FAB M6 phenotype [23,24]. Detailed studies reveal that CCL2 and CCL13 production varies among patients and correlates with the expression of other chemokines, such as CCL3, CCL4, CXCL1, and CXCL8/IL-8 for CCL2, and CCL17, CCL22, CCL24, and CXCL5 for CCL3 [31]. This indicates that CCR2 ligands may have similar roles in AML but only a particular MCP may be relevant to tumorigenesis in individual patients.

AML cells may increase CCL7 and CCL8 production under hypoxia conditions, making them a source of CCR2 ligands in AML patients’ bone marrow [43]. MSCs may also produce CCR2 ligands, with in vitro studies showing AML cells can increase CCL2 expression in MSCs [45], though this is not always confirmed [77]. Some studies suggest AML patients have lower CCL2 expression in MSCs compared to healthy individuals [78]. Additionally, mutations in the *IDH1* or *IDH2* genes result in R-2-hydroxyglutarate production, which increases CCL2 expression in bone marrow stromal cells [79].

The serum level of CCL2 in MDS patients is comparable to that of healthy individuals [41]. However, CCL2 expression is significantly higher in bone marrow CD34^+^ cells of MDS patients compared to healthy controls [42], and CCL2 levels are elevated in the bone marrow plasma of MDS patients as well [80]. Importantly, CCL2 levels are not linked to the responsiveness of MDS patients to 5-azacytidine chemotherapy [80], indicating that CCL2 may not have an anti-apoptotic effect on MDS cells.

#### 3.2.2. Action of CCR2 Ligands in AML

CCR2 ligands can directly affect AML cells. CCL2, CCL7, CCL8, and CCL13 increase AML cell proliferation in some patients [31], though CCL2 can also inhibit proliferation in a small group of patients. In most cases, these chemokines do not affect AML cell proliferation [31]. Studies on cell lines like HL-60, KG-1, THP-1, and U-937 show no effect of CCL2 on AML proliferation [81], although CCL2 can autocrinally increase proliferation in some cell lines, such as HL-60 [82]. CCR2 ligands can induce AML cell migration but do not affect AML cell mobilization from bone marrow [30,72].

CCR2 ligands do not directly affect AML cell chemoresistance, as CCL2 does not impact AML cell survival when treated with cytarabine [72]. However, CCL2 can induce senescence in MSCs, leading to a senescence-associated secretory phenotype (SASP), which could explain AML cells’ influence on chemokine expression in MSCs [45,76,83].

As a monocyte chemoattractant, CCL2 can recruit monocytes to AML cells, especially those with the FAB M4–M5 phenotypes [81]. These monocytes can reduce AML cell proliferation. The migration of monocytes to AML cells may explain the elevated number of macrophages observed in the bone marrow of AML patients [84]. In the AML bone marrow microenvironment, these macrophages undergo polarization into M2 macrophages [85], which exhibit immunosuppressive properties and inhibit the immune response. This, in turn, reduces the effectiveness of immunotherapy. Additionally, macrophages are linked to drug resistance in AML cells and are associated with poorer patient prognosis [85].

In MDS patients, CCL2 expression is significantly higher in bone marrow CD34^+^ cells, and the level of this chemokine in the bone marrow is elevated compared to healthy individuals [42,80]. CCL2 may play a role in recruiting myeloid-derived suppressor cells (MDSCs) to the bone marrow [86], and the number of MDSCs is increased in the bone marrow of MDS patients.

CCR2 ligands may have other clinical significance in AML (Table 2). AML patients often have reduced numbers of CD4^+^ and CD8^+^ T cells in their blood [87], as seen in the C1498 AML mouse model [88]. In mice, CCL2 levels in the thymus contribute to this process, with AML cells or other cells in the thymus being potential sources of this chemokine. However, patient studies are needed to confirm CCR2 ligands’ involvement in these clinical aspects of AML.

### 3.3. Ligands of the CCR3 Receptor

The ligands of CCR3 (CD193) that primarily act through this receptor are CCL11 (eotaxin-1), CCL24 (eotaxin-2, MPIF-2), and CCL26 (eotaxin-3, IMAC) [16,34]. Additionally, CCL11 can also activate CCR5 [16]. CCR3 is expressed on eosinophils, making the chemoattraction of these cells the most significant feature of CCR3 ligands, and thus, their participation in allergic reactions and diseases. CCR3 is also expressed on basophils, allowing CCR3 ligands to cause tissue infiltration by these cells [89]. Although CCR3 ligands are involved in allergic diseases, they do not seem to play a significant role in tumor mechanisms in AML.

#### Expression and Action of CCR3 Ligands in AML

The levels of CCL24 in the blood of AML patients do not differ from those in healthy individuals [39]. CCL24 expression in AML cells is lower when RAS proteins are activated in these cells [23,24]. AML cells do not secrete CCL11, and only half of the patients with AML have cells capable of producing CCL24. Approximately 20% of patients have AML cells that produce very small quantities of CCL26 [31]. CCL24 expression may correlate with other chemokines, particularly CCL13, CCL17, CCL22, CXCL5, and CXCL6 [31].

The serum level of CCL11 in MDS patients does not differ from that of healthy individuals [41]. In high-risk MDS patients the serum level of CCL11 is lower when In low-Risk MDS patients [41].

CCR3 expression in AML cells is extremely low compared to other chemokine receptors such as CCR1, CCR2, CCR4, CXCR2, and CXCR4 [31]. The expression level of CCR3 in AML is not linked to patient prognosis [23,24]. There is no significant information available about the effect of CCR3 ligands on tumorigenesis in AML, though it is known that CCR3 ligands can increase AML cell proliferation in some patients [31].

The collected information indicates that the CCR3 axis has no clinical significance in AML.

### 3.4. Ligands of the CCR4 Receptor

Among the ligands of CCR4 (CD194) are CCL17, also known as thymus and activation regulated chemokine (TARC), and CCL22, also known as macrophage-derived chemokine (MDC) [16,34]. Another CCR4 ligand, which is not a chemokine, is the chemokine-like factor (CKLF) [90]. CCR4 is expressed in activated CD8^+^ T cells, T_reg_, natural killer T (NKT) cells, and memory CD4^+^ T cells [16]. Therefore, the CCR4 axis plays an important role in immune system function.

This axis is also significant in AML, as evidenced by the association of the expression of its components with the prognosis of AML patients [23,24,91]. However, the molecular function of this axis in AML tumorigenesis requires further investigation.

#### 3.4.1. Expression of CCR4 and Its Ligands in AML

The level of CCL17 in the blood of adult AML patients is lower than in healthy individuals [92]. CCL17 is secreted by AML cells from only half of the patients, and the amounts are very low [31,92]. The expression level of CCL17 in AML cells does not depend on FAB classification or *FLT3* gene mutation status [92].

In contrast, CCL22 is secreted by AML cells in significantly higher amounts—10 times higher than CCL17 [31]. This suggests that CCL22 may play a more crucial role in AML tumorigenic processes. AML cells secrete CCL22 at much higher levels than bone marrow CD34^+^ cells and blood monocytes, which do not secrete detectable amounts of this chemokine [30]. Detectable secretion of CCL22 by AML cells occurs in only half of AML cases [31]. The level of CCL22 expression varies with FAB classification, being higher in AML cells with the FAB M4–M5 phenotypes compared to the less differentiated FAB M0–M2 subtypes [23,24,30]. The expression of CCL22 does not depend on *FLT3* gene mutations. Higher CCL22 expression in AML is associated with a worse prognosis for patients [23,24,91].

The production of CCL17 may correlate with the expression of both CCL17 and CCL22 in AML cells. The production of these chemokines may also correlate with the expression of other chemokines, particularly CCL13, CCL24, CXCL5, and CXCL6 [31], indicating the involvement of CCR2 and CXCR2 ligands, as well as CCR4 ligands, in tumorigenic mechanisms in certain AML patients.

CCR4 expression in AML cells is high compared to other chemokine receptors such as CCR3, CCR5, CXCR1, or CXCR2 [31]. The expression level of CCR4 is higher in AML cells with the FAB M0 phenotype than on those with the FAB M5 phenotype [23,24]. Moreover, a mutation in the *FLT3* gene is associated with lower CCR4 expression in AML cells. CCR4 expression is higher in CD34^+^ AML cells than in CD34^−^ AML cells [31], indicating that CCR4 ligands may be significant in leukemic stem cells. Higher CCR4 expression is associated with a trend (*p* = 0.06) towards a poorer prognosis for AML patients [27].

#### 3.4.2. Action of CCR4 and Its Ligands in AML

Elevated CCL22 expression in AML cells is linked to a poorer prognosis for patients [23,24,91]. Similarly, higher expression of the corresponding receptor in AML cells shows a trend (*p* = 0.06) towards worse outcomes for AML patients [27]. Currently, there are no studies on the significance of CKLF in AML. Bioinformatics analyses on the GEPIA portal (http://gepia.cancer-pku.cn accessed 3 July 2023) indicate that higher CKLF expression in AML cells is associated with a worse prognosis [27]. This suggests that CKLF may play a crucial role in AML tumorigenesis, though further investigation is needed. The correlation of higher chemokine and ligand expression with a poorer prognosis highlights the significant influence of the CCL22/CKLF–CCR4 axis on AML tumorigenesis. Therefore, the relevance of this axis in AML requires further investigation (Table 3).

It is known that CCR4 ligands can increase AML cell proliferation in a small number of patients [31]. Thus, the described axis in cancer processes may not be directly involved in acting on AML cells.

The activity of the CCL22/CKLF–CCR4 axis is likely centered in the bone marrow, associated with interactions between non-leukemic stromal cells and AML cells, resulting in increased production of CCL17 and other chemokines such as CCL5 and CXCL10 [92]. The described axis may also recruit T cells and NKT cells to the AML bone marrow microenvironment [16]. In solid tumors, the CCL22/CCL17–CCR4 axis is responsible for recruiting T_reg_ to the tumor niche [93]. Similarly, this axis facilitates Treg recruitment in various non-cancerous conditions [94,95]. This suggests that in AML, elevated CCL22 production in the bone marrow could lead to T_reg_ recruitment and accumulation, ultimately worsening the patient’s condition [46,60]. This connection between increased CCL22 expression in AML cells and poorer prognosis is well established [23,24,91].

Another possible explanation for the link between high CCL22 expression and poor prognosis is the recruitment of Th17 cells. Th17 cells express the CCR6 receptor, and as shown in studies on solid tumors, they can be recruited through this axis [96]. In the AML bone marrow microenvironment, Th17 cells play a role in cancer progression, including promoting chemoresistance [97]. Furthermore, the association between elevated CCL22 and CCR4 expression in AML cells and worse prognosis may be due to the facilitation of AML cell migration via this axis. However, the importance of CCL22 in the bone marrow microenvironment for AML has not yet been established.

### 3.5. Ligands of the CCR5 Receptor

CCR5 (CD195) is a receptor for several chemokines, including CCL3, CCL4 (also known as MIP-1β), CCL5 (also known as regulated on activation, normally T cell expressed and secreted (RANTES)), CCL7, CCL11, CCL14, and CCL16 [16]. Since CCR5 is primarily responsible for the properties of CCL4 and CCL5, these chemokines are reviewed in the context of this receptor. Notably, CCL4 and CCL5 also activate CCR1 and CCR3 [16]. CCR5 is expressed on activated CD8^+^ T cells, γδ T cells, natural killer (NK) cells, monocytes, macrophages, and DC [16]. Therefore, CCR5-activating chemokines play a crucial role in the proper immune response to pathogens [98] and the anti-cancer response in solid tumors [99].

#### 3.5.1. Expression of CCR5 and Its Ligands in AML

CCR5 ligands may play an important role in AML. Higher expression of CCR5 [32], CCL4 [27,39], and CCL5 [100,101] in AML cells is associated with a worse prognosis for AML patients. Similarly, higher levels of CCL5 in the blood of adult AML patients are associated with a worse prognosis [39]. Data from UALCAN indicate that increased expression of CCL5 and CCR5 is associated with a trend towards poorer prognosis for AML patients (CCR5 *p* = 0.071; CCL5 *p* = 0.068) [23,24].

Adult AML patients exhibit higher levels of CCL4 and CCL5 in their blood compared to healthy individuals [38]. After chemotherapy, these chemokine levels decrease, but they remain higher than in healthy individuals. Another study, however, shows that blood levels of CCL5 in adult AML patients are lower than in healthy individuals [39]. Higher CCL5 levels are observed in adult patients under 50 compared to those over 50 years old. Moreover, AML patients with lower blast proportions (<80%) in the peripheral blood have higher CCL5 levels than those with higher levels of AML cells [74]. Additionally, higher blood levels of CCL5 in adult AML patients may be associated with a higher likelihood of GVHD after bone marrow transplantation [74].

In MDS patients, serum levels of CCL5 are lower than in healthy individuals [41,102], with high-risk MDS showing even lower levels compared to low-risk MDS [41]. Conversely, CCL4 levels are similar between MDS patients and healthy controls [41]. However, in bone marrow CD34^+^ cells of MDS patients, CCL4 and CCR5 expression is elevated, while CCL5 expression is reduced [42]. There is also increased CCR1 expression, suggesting a potential autocrine effect of CCL4.

In the majority of adult patients, AML cells produce CCL4 and CCL5 [30,31], with CCL4 production being much higher than CCL5 [31]. CCL5 expression is highest in AML cells with FAB M5 and M7 phenotypes compared to FAB M0–M3 phenotypes [23,24,44], while CCL4 expression is highest in AML cells with the FAB M7 phenotype [23,24]. CCL4 expression is lower in AML cells with *FLT3* gene mutations compared to those without the mutation [23,24]. The expression level of CCL5 in AML cells is not significantly associated with *FLT3* mutation status. Another factor increasing CCL4 production in AML cells is MEIS1, which also increases the production of CCL3 and CXCL4 [48]. MEIS1 protein is often implicated in leukemogenesis in AML [49].

The expression of CCL4 and CCL5 chemokines may be slightly intercorrelated [31]. High production of CCL4 is better correlated with high production of CCL2, CCL3, CXCL1, and CXCL8 in AML cells, while CCL5 production is associated with CXCL9, CXCL10, CXCL11, and CCL23 [31,92].

CCR5 expression is higher in AML cells with the FAB M4–M5 phenotypes compared to those with the FAB M1–M3 phenotypes [23,24,30]. The level of CCR5 expression in AML cells is not significantly correlated with *FLT3* mutation status [23,24]. CCR5 receptor expression is higher in CD34^+^ AML cells than in CD34^−^ AML cells [31], indicating some importance of CCR5 and its ligands for AML leukemic stem cells.

Currently, there are no studies indicating higher levels of CCL4 or CCL5 in the bone marrow of AML patients compared to healthy individuals [40]. AML patients may have locally higher production of these chemokines in the bone marrow, which may be attributed to MSCs. In co-cultures of AML cells with MSCs, there is an increase in the production of CCL4 and CCL5 [45]. Hypoxia may also increase CCL4 and CCL5 production in the bone marrow by elevating their expression in AML cells [43].

#### 3.5.2. Action of CCR5 and Its Ligands in AML

The expression levels of CCR5, CCL4, and CCL5 in AML cells are closely correlated with the prognosis for AML patients [27,32,39,100,101], indicating that the CCL4/CCL5–CCR5 axis is important in AML tumorigenesis (Table 4).

The impact on prognosis may be attributed to the increase in AML cell proliferation caused by this axis. However, CCR5 ligands, CCL4, and CCL5 can only increase AML cell proliferation in some patients [31]. In most patients, they do not affect AML cell proliferation.

Chemoresistance may be another factor affecting prognosis. CCL5 and its ability to activate the CCR5 receptor cause resistance of AML with *FLT3* mutations to FLT3 tyrosine kinase inhibitors [103]. This is associated with the co-localization of CCR5 and CXCR4, where CCR5 ligands affect CXCR4 signal transduction. Treatment with FLT3 tyrosine kinase inhibitors may be followed by AML relapse, with AML cells showing increased production of CCL5 and consequently increased resistance to drug therapy. This resistance mechanism functions independently of *FLT3* gene mutations [103]. In vitro studies show that using CCL5-neutralizing antibodies can abolish the resistance of AML cells with *FLT3* mutations to FLT3 tyrosine kinase inhibitors, suggesting CCR5 or its ligands as convenient therapeutic targets. However, no in vivo studies in laboratory animals or human clinical trials of CCR5 inhibitors to enhance current AML therapy efficacy have been conducted yet.

CCR5 ligands may also participate in other tumorigenic processes in AML, which correlates with prognosis. These ligands may induce T_reg_ recruitment to bone marrow in a CCR5-dependent manner [46], preventing the immune system from responding to AML. CCL3 is responsible for T_reg_ recruitment, as indicated in in vivo mouse model experiments. The action of CCL3 may depend on CCR1 and CCR5 receptors [46]. Other chemokine axes, particularly CXCL12–CXCR4, also play a significant role in T_reg_ recruitment to bone marrow [46,57]. The CXCL12–CXCR4 and CCL3–CCR1/CCR5 axes may be crucial in this process in AML patients [57]. T_reg_ possess immunosuppressive properties [46,60], weakening the immune system’s response against AML. This, in turn, contributes to a deterioration in the patient’s condition.

CCR5 may also be important in developing extramedullary leukemia of the skin in pediatric AML patients [62]. Specifically, CCL3 may be responsible, as its expression occurs in extramedullary leukemia of the skin, while CCL5 expression is absent, indicating that CCL5 may not play a role in this aspect of AML. Patients with extramedullary leukemia of the skin show no difference in CCR5 expression levels in AML cells in peripheral blood compared to healthy subjects [62]. Further studies are required to confirm the importance of CCR5 and its ligands in forming extramedullary leukemia of the skin in pediatric AML patients.

In MDS patients, CCL5 may have an anti-tumor function. Serum levels of CCL5 are lower in these patients compared to healthy individuals [41,102]. CCL5 plays a key role in recruiting cytotoxic lymphocytes, contributing to the immune response, including the defense against solid tumors [16,98,99]. The reduced CCL5 levels may indicate immune system dysfunction in MDS patients. Additionally, in the bone marrow CD34^+^ cells of MDS patients, expression of CCL4, CCR5, and CCR1 is elevated compared to healthy controls [42], suggesting an autocrine action of CCL4. This chemokine, produced by MDS cells, acts directly on these cells.

### 3.6. Ligands of the CCR6 Receptor

CCL20, also known as liver and activation-regulated chemokine (LARC), MIP-3α, and exodus-1, is a ligand for CCR6 (CD196) [16,34]. This receptor is expressed in B cells, memory CD4^+^ T cells, basophils, and T_reg_ cells [16]. The importance of the CCL20–CCR6 axis in AML tumor mechanisms has not been extensively studied. However, public database searches suggest that the CCL20–CCR6 axis may play a significant role in tumorigenesis and could be an interesting therapeutic target in AML [23,24,27].

#### 3.6.1. Expression of CCR6 and Its Ligands in AML

In adult patients, CCR6 expression is higher in AML cells with the FAB M4–M5 phenotypes than in those with the FAB M0–M1 phenotypes [30]. Additionally, CCR6 expression is higher in AML with the *FLT3* mutation compared to AML without this mutation [23,24]. AML cells with the FAB M4–M5 phenotypes produce higher levels of CCL20 compared to monocytes and bone marrow CD34^+^ cells [30]. In contrast, less differentiated AML cells produce CCL20 in lower amounts and only in half of the cases [30]. CCL20 can increase AML cell proliferation, but only in some patients [31]. In most patients, it does not affect AML cell proliferation.

Both serum and bone marrow plasma levels of CCL20 are elevated in MDS patients compared to healthy individuals [104]. This increase is also observed in bone marrow cells. In the bone marrow mononuclear cells of MDS patients, there is higher expression of CCL20 and its receptor, CCR6, than in healthy controls [104]. Specifically, CCL20 expression is significantly elevated in the bone marrow CD34^+^ cells of MDS patients compared to healthy individuals [42].

#### 3.6.2. Action of CCR6 and Its Ligands in AML

Analysis conducted on the GEPIA portal (http://gepia.cancer-pku.cn accessed 5 July 2023) indicates that higher CCR6 expression in AML cells is associated with a worse prognosis for patients [27]. In contrast, analysis on UALCAN (https://ualcan.path.uab.edu accessed 5 July 2023) suggests that higher CCL20 expression in AML cells is associated with a better prognosis [23,24]. This indicates that the CCL20–CCR6 axis may have two clinically relevant functions: while CCL20 secretion by AML cells might have an anti-tumor effect, the CCR6 receptor could have a pro-tumor effect in AML.

No studies are currently available on the anti-leukemic properties of CCL20 in AML, but it can be speculated that CCL20 secreted by AML cells might promote the migration of cytotoxic lymphocytes, which have anti-tumor effects [105]. Another clinically relevant component of the CCL20–CCR6 axis is the action of CCR6 in AML cells, making it an interesting therapeutic target. However, further studies are needed to understand the exact involvement of CCR6 in AML cells.

In MDS patients, elevated expression of CCL20 and CCR6 in the bone marrow suggests activation of the CCL20–CCR6 axis [104]. However, the exact role of this axis in MDS has not been thoroughly investigated.

### 3.7. Ligands of the CCR7 Receptor

There are two chemokines that are ligands of CCR7 (CD197): CCL19, also known as EBI1-ligand chemokine (ELC), MIP-3β, CKb11, exodus-3, and CCL21, also known as secondary lymphoid tissue chemokine (SLC), 6Ckine, exodus-2 [16,34]. CCR7 expression is highest in T cells [16], indicating that this axis is particularly important for these cells, especially in lymph nodes. Collected data suggest that this axis may not have a significant role in AML [23,24,27].

In the bone marrow CD34^+^ cells of MDS patients, CCR7 expression is higher compared to healthy individuals [42]. However, the role of CCR7 in MDS remains unclear.

#### 3.7.1. Expression of CCR7 and Its Ligands in AML

CCR7 expression in AML cells is higher than in cord blood-derived CD34^+^ cells [29]. However, AML cells have low expression of CCR7 compared to other chemokine receptors [30]. The highest expression of CCR7 is observed in AML cells with the FAB M6 phenotype [23,24]. Additionally, CCR7 expression is lower in AML cells with *FLT3* mutations compared to those without the mutation [23,24].

CCL19 production in AML cells occurs only in some patients and only in cells with the FAB M4–M5 phenotypes [30]. This chemokine is not produced by monocytes, bone marrow CD34^+^ cells, or AML cells with the FAB M0–M1 phenotype. In contrast, CCL21 is not produced by AML cells in detectable amounts [31].

#### 3.7.2. Action of CCR7 and Its Ligands in AML

Bioinformatic analyses performed using the GEPIA and UALCAN databases do not indicate that the expression level of CCR7 or its ligands is associated with prognosis in AML [23,24,27]. This suggests that the CCL19/CCL21–CCR7 axis may not be significantly important for tumorigenic mechanisms in AML.

### 3.8. Ligands of the CCR8 Receptor

CCR8 (CDw198) ligands include CCL1, also known as I-309, and CCL18, also known as MIP-4 or pulmonary and activation-regulated chemokine (PARC) [16,34]. This classification is somewhat artificial, as it appears that the primary receptor for CCL18 is phosphatidylinositol Transfer Protein, Membrane-Associated 3 (PITPNM3), also known as NIR1 or ACKR6 [106]. CCL18 also activates CCR3, CCR8, and G-protein-coupled estrogen receptor 1 (GPER1) [107,108]. CCL18 is a marker of M2 macrophage polarization [109]. In solid tumors, CCL18 is produced almost exclusively by tumor-associated macrophages (TAM) [110].

CCR1 is expressed in T_reg_ cells, memory CD4^+^ T cells, and activated CD8^+^ T cells [16]. The role of CCR8 in AML tumorigenesis is not well established and it likely does not play a significant role. The level of CCR8 expression in AML cells is not associated with prognosis in patients with AML [23,24].

#### 3.8.1. CCL1 in AML

The CCL1 chemokine is produced in detectable amounts by AML cells in two-thirds of patients [31]. However, it may not significantly impact AML cell proliferation. In some cases, it can either increase or decrease AML cell proliferation, depending on the individual patient, as shown in in vitro studies [31]. The level of CCL1 expression in AML cells is not correlated with patient prognosis [23,24], indicating that this chemokine is not significant in the course of AML.

#### 3.8.2. CCL18 in AML

The importance of CCL18 in AML tumorigenesis has not been well-researched. Adult AML patients have higher levels of CCL18 in their bone marrow compared to healthy individuals [40]. However, in pediatric AML patients, blood levels of CCL18 do not differ from those of healthy individuals [69]. CCL18 may be produced in the highest amounts by AML cells with the FAB M6 phenotype [23,24]. CCL18 expression is lower in AML cells with the *FLT3* mutation compared to those without this mutation [23,24]. The level of CCL18 expression in AML is not associated with patient prognosis [23,24].

PITPNM3 receptor expression is highest in AML cells with the FAB M0 phenotype and lowest in AML cells with the FAB M5 phenotype [23,24]. PITPNM3 expression is also lower in AML cells with the *FLT3* mutation. However, the level of PITPNM3 expression in AML cells does not affect patient prognosis [23,24].

CCL18 increases the proliferation of AML cells in a small number of adult patients but does not affect proliferation in most cases [31]. In AML patients, the source of CCL18 in the bone marrow is likely macrophages, not AML cells. AML patients exhibit an accumulation of macrophages in the bone marrow [84]. In solid tumors, these cells are the main source of CCL18 [110]. A similar relationship may occur in AML, but further investigation is required.

### 3.9. Ligands of the CCR9 Receptor

CCL25, also known as thymus-expressed chemokine (TECK), is the only ligand for CCR9 (CD199) [16]. This chemokine is predominantly expressed in the small intestinal epithelium and thymus, where it plays an immunological role [111]. CCL25 acts on B cells, DC, γδ T cells, NK cells, and NKT cells [16,111].

#### 3.9.1. Expression of CCR9 and Its Ligands in AML

Higher expression of CCL25 in AML cells is associated with a poorer prognosis for AML patients, indicating the significant relevance of the CCL25–CCR9 axis in AML tumorigenesis [23,24]. The highest expression of CCL25 is found in AML cells with the FAB M7 phenotype, while the lowest expression is in AML cells with the FAB M3 phenotype [23,24]. However, another study does not reveal any detectable production of CCL25 in AML cells [31]. CCL25 expression is lower in AML cells with the *FLT3* gene mutation [23,24].

Nevertheless, the importance of the CCL25–CCR9 axis in AML tumorigenesis has not yet been thoroughly researched. CCR9 expression is found in AML cells [30], but there are no differences in its expression between AML cells with the FAB M4–M5 phenotypes and those with the FAB M0–M1 phenotypes [23,24,30]. However, CCR9 expression in AML cells with the FAB M4–M5 phenotypes may be higher than in monocytes [30]. Additionally, CCR9 expression is lower in AML cells with the *FLT3* mutation compared to those without the mutation (Table 5) [23,24].

#### 3.9.2. Action of CCR9 and Its Ligands in AML

The function of CCR9 in AML cells has not been extensively studied. It is known that CCL25 can increase AML cell proliferation in some patients [31], but in most patients, this chemokine does not affect AML cell proliferation. Further research is needed to fully understand the role of the CCL25–CCR9 axis in AML.

### 3.10. Ligands of the CCR10 Receptor

The ligands of CCR10 are CCL27, also known as cutaneous T-cell-attracting chemokine (CTACK) or ESkine, and CCL28, also known as mucosa-associated epithelial chemokine (MEC) [16,28,34]. These chemokines play physiological roles in regulating the immune system in the skin (CCL27) and mucosal tissues (CCL28) [112]. Additionally, CCL28 functions as a growth factor for primitive hematopoietic cells [113], suggesting that CCL28 could have significance in tumorigenic processes in AML within the bone marrow. Further research is needed to fully understand the roles of CCL27 and CCL28 in AML tumorigenesis, particularly how CCL28 might influence AML in the bone marrow environment.

#### 3.10.1. Expression of CCR10 and Its Ligands in AML

Bioinformatic analysis from the UALCAN portal (https://ualcan.path.uab.edu accessed 21 June 2023) indicates that higher CCR10 expression in AML cells is associated with a poorer prognosis for patients [23,24]. This suggests that the CCL27/CCL28–CCR10 axis plays an important role in AML tumorigenesis and could be a potential therapeutic target. However, this axis has not been well studied.

It is known that AML cells only secrete CCL28 in some patients [31,114]. CCL28 expression is lowest in AML cells with the FAB M3 and M5 phenotypes [23,24]. Both CCR10 and CCL28 expressions are lower in AML cells with *FLT3* gene mutations [23,24]. In AML FAB M3, the presence of the fusion protein PML-RARα is associated with lower expression levels of CCL28 [23,24]. The expression of CCL28 is not associated with the expression of other chemokines [114].

#### 3.10.2. Action of CCR10 and Its Ligands in AML

In some patients, CCR10 ligands can increase AML cell proliferation, while in a small number of patients, they can decrease proliferation [31,114]. It appears that CCR10 ligands modulate the effects of hematopoietic growth factors such as granulocyte-macrophage colony-stimulating factor (GM-CSF), stem cell factor (SCF), and FMS-like tyrosine kinase 3 ligand (FLT3L) on AML cells [114]. CCR10 expression is lowest in AML cells with the FAB M3 phenotype and highest in AML cells with the FAB M7 phenotype (Table 6) [23,24].

Although CCR10 ligands perform physiological functions in the skin and mucosal tissues [112], this axis does not seem to be associated with extramedullary leukemia of the skin, at least in pediatric AML patients [62].

## 4. γ-Chemokines

γ-chemokines are another subfamily of chemokines, including XCL1 (also known as single C motif (SCM)-1α, activation-induced T-cell-derived and chemokine-related molecule (ATAC)) and XCL2 (also known as SCM-1β) [28,34]. XCR1 is the receptor for these chemokines [16,115]. γ-chemokines play an important role in the functioning of DC [115].

The significance of XCL1 and XCL2 in AML tumorigenesis has been poorly studied (Table 7). An extensive bioinformatic analysis found that higher levels of XCR1 expression in AML cells in adult patients are associated with a poorer prognosis [116]. This indicates an important clinical function for this receptor in AML cells. This finding may be attributed to the effect of XCR1 ligands on AML proliferation. XCL1 can increase AML cell proliferation in some patients [31]. However, in most patients, this chemokine does not affect proliferation. Expression levels of XCL1 and XCL2 in AML cells are low [23,24]. According to FAB classification, the highest expression of these chemokines is found in AML cells with the FAB M7 phenotype [23,24]. Furthermore, the expression of XCL1 and XCL2 is lower in AML with mutations in the *FLT3* gene [23,24].

## 5. δ-Chemokines

δ-chemokines with a CX3C motif at the N-terminus form the final sub-family of chemokines, with CX3CL1 (also known as fractalkine) being the sole member. The receptor for this chemokine is CX3CR1. CX3CL1 is synthesized as a trans-membrane protein in the cell membrane, where it can function as an adhesion protein, facilitating the attachment and transmigration of leukocytes by binding to CX3CR1 [18,117,118,119,120]. The soluble form of CX3CL1, released through proteolytic digestion by proteinases such as ADAM17 and MMP2, can also activate the CX3CR1 receptor [117,121].

### 5.1. Expression of CX3CR1 and Its Ligand in AML

The CX3CL1–CX3CR1 axis may play a crucial role in AML tumorigenesis, but its significance has not been thoroughly investigated. Higher expression of CX3CR1 is found in AML cells compared to cord blood CD34^+^ cells [29,122]. The highest expression of this receptor is observed in AML cells with the FAB M4–M5 phenotypes [23,24]. In pediatric AML patients, CX3CR1 expression in AML cells may vary depending on specific disease characteristics, with lower expression in AML cells and a lower percentage of blasts in the blood and bone marrow [123]. Additionally, AML cells with a mutation in the KMT2A/MLL gene exhibit higher CX3CR1 expression, while mutations in *FLT3* and *NPM1* do not appear to affect CX3CR1 expression levels [123].

CX3CL1 expression is partially influenced by the FAB classification, with AML cells having the lowest expression of this chemokine in the FAB M0 phenotype [23,24]. Elevated CX3CL1 levels in the bone marrow of adult AML patients may suggest a pro-cancer effect of this axis in the bone marrow, although specific studies on this topic are still lacking [40].

### 5.2. Action of CX3CR1 and Its Ligand in AML

The CX3CL1–CX3CR1 axis may be clinically important, as higher CX3CR1 expression in AML cells is associated with a worse prognosis in both pediatric patients with hyperleukocytosis and adult AML patients [32,123]. Data from UALCAN also indicate a trend towards poorer prognosis with higher CX3CR1 expression (*p* = 0.051) [23,24].

The association of CX3CR1 expression with patient prognosis may be related to CNS leukemia. Higher CX3CR1 expression in AML cells in pediatric patients is linked to a higher likelihood of AML cells infiltrating the CNS [123]. Given that CX3CL1 expression is highest in the heart and brain [124], the invasion of AML cells into the brain might depend on the CX3CL1–CX3CR1 axis. Experiments on U-937 cells show that CX3CR1 expression increases when CXCR4 activity is blocked by an inhibitor, suggesting a potential side effect of CXCR4-targeting drugs, where AML cells might migrate to the CNS after release from the bone marrow [125].

Another possible reason for the link between CX3CR1 expression and patient prognosis is the effect of this chemokine on AML cell proliferation. CX3CL1 may influence AML cell proliferation, especially in the CNS where its expression is high [124]. CX3CL1 can elevate AML cell proliferation in some patients, although in most cases, it likely does not affect leukemic cell proliferation significantly [31].

## 6. Atypical Chemokine Receptors

Atypical chemokine receptors are chemokine receptors that, by definition, do not cause cell migration [126]. They are involved in regulating chemokines by transporting them to other areas of the extracellular space [127], retaining chemokines on the cell surface and regulating chemokine bioavailability [128,129], chemokine degradation [130], or regulating chemokine receptors [131]. Several representatives belong to this family, and there are also a few membrane proteins suspected of binding chemokines but not yet classified as atypical chemokine receptors [132]. These include:ACKR1 (Duffy antigen receptor for chemokines, DARC, CD234): Found on erythrocytes and in blood vessels, ACKR1 is responsible for the bioavailability of chemokines, binding chemokines of the CXC (α-chemokines) and CC (β-chemokines) subfamilies [16,34,128,129].ACKR2 (decoy receptor D6): Responsible for the degradation of chemokines, ACKR2 silences inflammatory reactions by binding chemokines of the CC subfamily (β-chemokines) [16,130].ACKR3 (CXCR7, CMKOR1): Exists in a hetero-dimer with CXCR4, regulating the action of CXCL12 by binding CXCL11 and CXCL12 [131,133]. Since this receptor primarily functions in α-chemokines, it is not discussed further here.ACKR4 (CCR11, chemokine receptor-like CCRL1, CCBP2, CCX-CKR): A receptor for CCL19, CCL20, CCL21, CCL22, and CCL25, ACKR4 is important in lymph node function [34,134,135].CCRL2 (ACKR5): A receptor for chemerin and possibly for CCL19, potentially modulating the CCL19–CCR7 axis [136,137,138,139]. It has not yet been officially included in the atypical chemokine receptor group due to the lack of confirmation of CCL19 binding [34,132].PITPNM3 (NIR1, ACKR6): A receptor for CCL18, responsible for the properties of this chemokine [106]. It is discussed in the context of CCL18 and CCR8.

In AML cells, there are no significant differences in the expression of ACKR1 and ACKR2 relative to controls, as indicated by analyses of the GEPIA website (http://gepia.cancer-pku.cn accessed 3 July 2023) [27]. Additionally, according to the UALCAN website (https://ualcan.path.uab.edu accessed 3 July 2023), the expression levels of ACKR1 and ACKR2 in AML cells are not associated with patient prognosis [23,24]. Furthermore, the expression of ACKR1 and ACKR2 in AML is lower in cells with *FLT3* mutations than in those without [23,24]. These data indicate that ACKR1 and ACKR2 do not play an important role in AML tumorigenesis and may not be significant in AML therapy.

Analyses performed on the GEPIA portal showed that ACKR4 expression in AML cells is low and not different from the control [27]. UALCAN analyses indicate that ACKR4 expression is highest in AML cells with the FAB M0-M2 phenotypes and lowest in those with the FAB M5 and M7 phenotypes [23,24]. ACKR4 expression in AML cells is lower in those with *FLT3* mutations. The expression level of this receptor is not associated with patient prognosis [23,24], suggesting that ACKR4 may not be significant in AML tumorigenesis or clinically relevant for this leukemia.

CCRL2 may have clinical significance in AML. Its expression in AML cells is higher than on cord blood-derived CD34^+^ cells [29]. The expression level of CCRL2 is not associated with *FLT3* gene mutations [23,24,29]. Higher CCRL2 expression in AML cells is associated with a poorer prognosis [91], and UALCAN analyses show a trend towards worse prognosis with higher CCRL2 expression (*p* = 0.091) [23,24]. The association of CCRL2 expression with patient prognosis may be due to its stimulation of AML cell proliferation and its role in increasing DNA methylation, leading to resistance of secondary AML cells to azacitidine [139]. Further studies on CCRL2’s significance in AML tumorigenesis are required.

These findings underscore the need for more research into the role of atypical chemokine receptors, particularly CCRL2, in AML tumorigenesis and potential therapeutic applications.

## 7. Conclusions

The importance of β-chemokines, γ-chemokines, and δ-chemokines in AML tumorigenesis has been poorly researched. Searches of public databases have indicated that the expression levels of certain chemokines and chemokine receptors in AML are closely associated with patient prognosis. This paper has highlighted such associations in the cases of the CCL22–CCR4 and CCL25–CCR9 axes, as well as for the CCR6, CCR10, XCR1, and CCRL2 receptors, suggesting that these chemokines and receptors could be potential therapeutic targets for anti-leukemia therapy. However, the precise role of these chemokines and chemokine receptors in AML remains undetermined. Therefore, it is currently unclear how these chemokines contribute to AML tumorigenesis and why elevated expression in AML cells is linked to poorer patient prognosis. Further research is necessary to elucidate these mechanisms and their implications for patient outcomes.

## Figures and Tables

**Figure 1 cancers-16-03246-f001:**
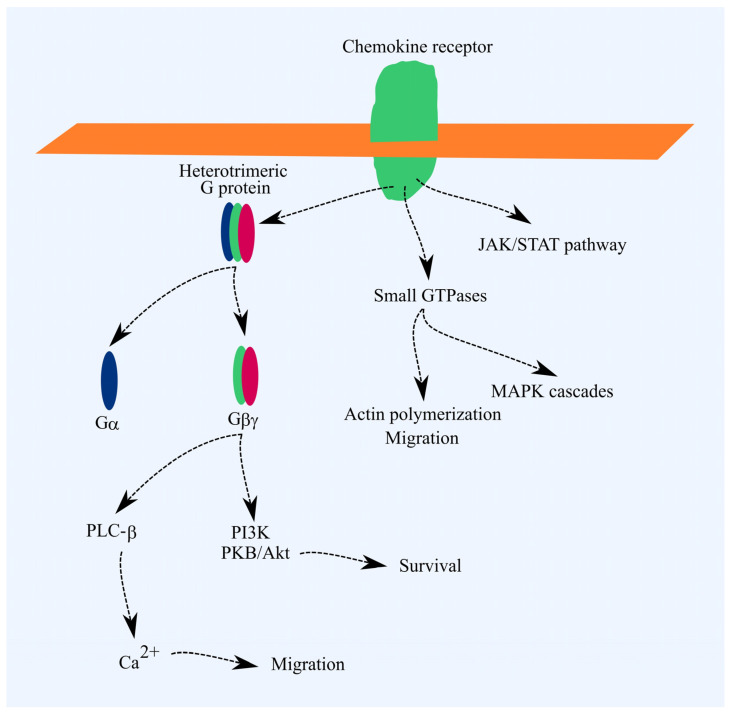
Chemokine receptor signal transduction. Ligand binding initiates the activation of heterotrimeric G proteins, which transmit signals to PLC-β and PI3K. PLC-β raises cytoplasmic Ca^2+^ levels, promoting cell migration, while Akt/PKB exerts anti-apoptotic effects. Chemokine receptors also activate small GTPases, leading to actin polymerization and MAPK cascade activation. Actin polymerization drives cell migration, and MAPK cascades support cell proliferation. However, these effects are not the primary outcomes of chemokine receptor activation.

**Figure 2 cancers-16-03246-f002:**
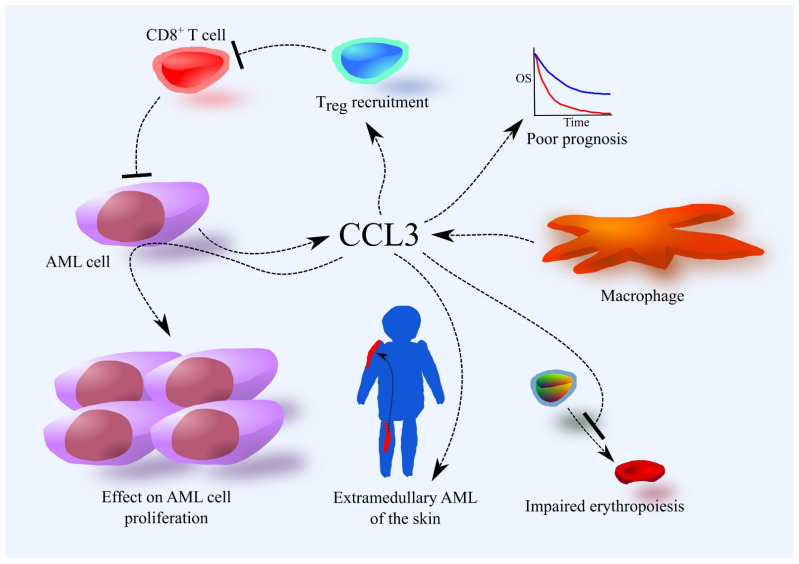
Importance of the CCL3 chemokine in AML. The source of CCL3 in the bone marrow are AML cells and macrophages. Higher expression of CCL3 in AML cells is associated with a poorer prognosis. This is related to the involvement of CCL3 in tumor processes in AML. CCL3 induces T_reg_ recruitment and accumulation in bone marrow, which suppresses the immune system’s fight against AML. CCL3 also affects the proliferation of AML cells. CCL3 also disrupts erythropoiesis. CCL3 is likely responsible for the development of extramedullary AML of the skin, at least in pediatric patients.

**Table 1 cancers-16-03246-t001:** The characteristics of the CCR1 receptor and its ligands in AML.

Protein	Expression in AML	Involvement in Tumorigenesis in AML	Sources
CCR1	Higher expression on AML cells compared to cord blood-derived CD34^+^ cells. The highest expression in AML cells with FAB M4–M5 phenotypes. In these cells, the expression level is similar to that in monocytes. Higher expression on CD34^+^ AML cells than on CD34^−^ AML cells.	Higher expression in AML cells signifies a worse prognosis.	[23,24,29,30,31,32]
CCL3	Higher levels in the blood and bone marrow of AML patients compared to healthy subjects. AML cells with the FAB M4–M5 phenotypes exhibit higher production than less differentiated AML cells. These AML cells also secrete more CCL3 than monocytes. Lower production in AML cells with the FAB M0–M1 phenotypes compared to CD34^+^ bone marrow cells.	Depending on the published paper, it can either decrease or increase the proliferation rate of AML cells. Causes recruitment and accumulation of T_reg_ in the bone marrow of AML patients. In pediatric patients, it is involved in the formation of extramedullary AML of the skin. Disrupts erythropoiesis leading to a decrease in the number of erythrocytes and platelets in the blood. Higher expression of CCL3 in AML cells is associated with a poorer prognosis	[23,24,30,31,36,37,38,39,46,53,54,55,56,57,62]
CCL14	Lower levels in AML cells with mutations in the *FLT3* gene.	In some patients, it may increase AML cell proliferation. No association between expression in AML cells and prognosis.	[23,24,31,69]
CCL15		In some patients, it may increase AML cell proliferation. No association between expression in AML cells and prognosis.	[23,24,31]
CCL16		In some patients, it may increase AML cell proliferation. No association between expression in AML cells and prognosis.	[23,24,31]
CCL23	Higher levels in the bone marrow of AML patients compared to healthy individuals. The highest expression in AML cells with FAB M4–M5 phenotypes.	A trend (*p* = 0.059) of higher expression in AML cells with worse prognosis. In some patients, it causes an increase, and in others, a decrease in AML cell proliferation. It disrupts erythropoiesis, leading to a decrease in the number of erythrocytes and platelets in the blood	[23,24,31,40,66,68]

**Table 2 cancers-16-03246-t002:** The characteristics of the CCR2 receptor and its ligands in AML.

Protein	Expression in AML	Involvement in Tumorigenesis in AML	Sources
CCR2	Higher expression in AML cells than in cord blood-derived CD34^+^ cells, but lower than in monocytes. The highest expression in AML cells with the FAB M4–M5 phenotype. The expression in AML cells with the FAB M4–M5 phenotypes is similar to that on monocytes. Higher expression on CD34^+^ AML cells than on CD34^−^ AML cells. Higher expression in AML cells with inv(16).	AML cell expression levels are not associated with prognosis.	[23,24,29,30,31]
CCL2	The lowest levels in blood in AML with FAB M4–M5 phenotypes. Lower levels in blood with lower levels of AML blasts in blood. Highest expression in AML cells with FAB M4–M5 phenotypes. Reduced expression in AML cells with *FLT3* gene mutation.	Expression level in AML cells is not associated with prognosis. Chemoattractant and recruiter of monocytes. Causes senescence of MSC cells in bone marrow. Thymic dysfunction in patients with AML	[23,24,30,31,45,73,74,76,81,83,88]
CCL7	Expression in AML cells with FAB M4–M5 phenotypes. No expression in AML cells with FAB M0–M1 phenotypes.		[23,24,30]
CCL8	Expression in AML cells with FAB M4–M5 phenotypes. No expression in AML cells with FAB M0–M1 phenotypes. The highest expression in AML cells with FAB M6 phenotype.	Expression level in AML cells is not associated with prognosis.	[23,24,30]
CCL13	The highest expression in AML cells with FAB M6 phenotype.	Expression level in AML cells is not associated with prognosis.	[23,24]

**Table 3 cancers-16-03246-t003:** The characteristics of the CCR4 receptor and its ligands in AML.

Protein	Expression in AML	Involvement in Tumorigenesis in AML	Sources
CCR4	The highest expression in AML cells with the FAB M0 phenotype. Lower expression in AML cells with *FLT3* gene mutation. Higher expression on CD34^+^ AML cells than on CD34^−^ AML cells.	Higher expression level in AML cells is associated with a trend (*p* = 0.06) of poorer prognosis.	[23,24,27,31]
CCL17	Lower blood levels in in AML patients.		[23,24,31,92]
CCL22	Higher production in AML cells compared to bone marrow CD34^+^ cells and blood monocytes. The highest expression in AML cells with FAB M4–M5 phenotypes.	Higher expression in AML cells signifies a poorer prognosis.	[23,24,30,31,91]
CKLF	The lowest expression in AML cells with the FAB M3 phenotype.	Higher expression in AML cells signifies a poorer prognosis.	[23,24,27]

**Table 4 cancers-16-03246-t004:** The characteristics of the CCR5 receptor and its ligands in AML.

Protein	Expression in AML	Involvement in Tumorigenesis in AML	Sources
CCR5	The highest expression in AML cells with the FAB M4–M5 phenotypes. Higher expression on CD34^+^ AML cells than on CD34^−^ AML cells.	Higher expression in AML cells signifies poorer prognosis. In AML with *FLT3* mutations, causing resistance to FLT3 tyrosine kinase inhibitors. Involved in T_reg_ recruitment to the bone marrow.	[23,24,30,31,32,46,57,103]
CCL4	The highest expression in AML cells with the FAB M7 phenotype. Lower expression in AML cells with mutation in the *FLT3* gene.	Higher expression in AML cells signifies a poorer prognosis.	[23,24,27,39]
CCL5	Higher levels in the blood of patients with a lower percentage of AML blasts in the blood. Highest expression in AML cells with FAB M5 and M7 phenotypes.	Higher expression in AML cells signifies a poorer prognosis. In AML with *FLT3* mutations, CCL5 causes resistance to FLT3 tyrosine kinase inhibitors.	[23,24,39,44,74,100,101,103]

**Table 5 cancers-16-03246-t005:** The characteristics of the CCR9 receptor and its ligands in AML.

Protein	Expression in AML	Involvement in Tumorigenesis in AML	Sources
CCR9	Higher expression in AML cells with FAB M4–M5 phenotypes than on monocytes. Lower expression in AML cells with *FLT3* gene mutation.	Expression level in AML cells is not associated with prognosis.	[23,24,30]
CCL25	The highest expression in AML cells with the FAB M7 phenotype, the lowest in AML cells with the FAB M3 phenotype. Lower expression in AML cells with *FLT3* gene mutation.	Higher expression in AML cells signifies a poorer prognosis.	[23,24,30]

**Table 6 cancers-16-03246-t006:** The characteristics of the CCR10 receptor and its ligands in AML.

Protein	Expression in AML	Involvement in Tumorigenesis in AML	Sources
CCR10	The highest expression is the highest in AML cells with the FAB M7 phenotype, and the lowest in AML cells with the FAB M3 phenotype. Lower expression in AML cells with *FLT3* gene mutation.	Higher expression in AML cells signifies a poorer prognosis.	[23,24]
CCL27			
CCL28	Expression is the lowest in AML cells with FAB M3 and M5 phenotypes. Lower expression is observed in AML cells with mutation in the *FLT3* gene.	Modulates the action of hematopoietic growth factors.	[23,24,114]

**Table 7 cancers-16-03246-t007:** The characteristics of the XCR1 receptor and its ligands in AML.

Protein	Expression in AML	Involvement in Tumorigenesis in AML	Sources
XCR1		Higher expression in AML cells signifies a poorer prognosis.	[94]
XCL1	The highest expression in AML cells with FAB M7 phenotype Lower expression in AML cells with *FLT3* gene mutation	Possibly increases the rate of AML cell proliferation.	[23,24,31]
XCL2	The highest expression in AML cells with the FAB M7 phenotype Lower expression in AML cells with mutation in the *FLT3* gene		[23,24]

## Data Availability

Not applicable.

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
