# Peer review of "Clinical Aspects and Significance of β-Chemokines, γ-Chemokines, and δ-Chemokines in Molecular Cancer Processes in Acute Myeloid Leukemia (AML) and Myelodysplastic Neoplasms (MDS)"

_cancers, 2024, doi:10.3390/cancers16193246_

Round 1

Reviewer 1 Report

Comments and Suggestions for Authors

The role of chemokines in acute myeloid leukemia is mostly unknown. In this paper the authors  provided a review of available literature data on chemokines and on their receptors in AML and on their association with prognosis.

The paper contains several information and underlines the need of a dedicated research  to clarify the real role of chemokines in  leukemia pathogenesis. 

I have just few questions:

·      Many chemokines seems to be overexpressed in AML with monocytic morphology. What can be the explanation? Pleas add a comment.

·      The authors considered only FAB classification and FLT3 mutation, but it would be interesting a correlation with other prognostic scores or other molecular/cytogenetic alteration (such as NPM, TP53, core binding factor mutations and so on). 

·      The importance of AML microenvironment has been emerged in determine response to therapy and prognosis. Did they find any relationship with immune composition? 

Author Response

Review 1 Answer

The role of chemokines in acute myeloid leukemia is mostly unknown. In this paper the authors  provided a review of available literature data on chemokines and on their receptors in AML and on their association with prognosis.

The paper contains several information and underlines the need of a dedicated research  to clarify the real role of chemokines in  leukemia pathogenesis. 

I have just few questions:

Comments 1:      Many chemokines seems to be overexpressed in AML with monocytic morphology. What can be the explanation? Pleas add a comment.

Response 1: The following fragment has been added:

AML cells with FAB M4-M5 phenotypes show elevated expression of several chemokines and chemokine receptors. This may be linked to epigenetic changes (e.g., DNA methylation) and increased expression and activation of various transcription factors in these cells compared to other AML phenotypes. For instance, NF-κB, a key transcription factor in inflammatory responses, is notably active. Another example is PU.1/SPI1, which is most highly expressed in AML cells with the FAB M5 phenotype. PU.1/SPI1 drives the expression of CCL22 in macrophages and dendritic cells (DC), potentially explaining the elevated CCL22 levels in AML cells with a monocytic phenotype. Nevertheless, the direct causes of increased chemokine and chemokine receptor expression in AML cells across different phenotypes are seldom explored experimentally.

Comments 2:      The authors considered only FAB classification and FLT3 mutation, but it would be interesting a correlation with other prognostic scores or other molecular/cytogenetic alteration (such as NPM, TP53, core binding factor mutations and so on). 

Response 2: The UALCAN database does not show a correlation between the expression of the discussed chemokines and the mutation status of NPM or TP53. However, it does reveal a link with the presence of the PML/RAR fusion protein and the activation of RAS proteins. We have incorporated this data into our article.

Comments 3:     The importance of AML microenvironment has been emerged in determine response to therapy and prognosis. Did they find any relationship with immune composition? 

Response 3: We agree with the reviewer’s assessment. Chemokines within the AML bone marrow microenvironment not only act directly on AML cells but also facilitate the recruitment of immune cells to the bone marrow. These recruited cells can either contribute to tumor progression or aid in the fight against AML. Therefore, we have expanded our review to cover the indirect roles of chemokines as well.

Reviewer 2 Report

Comments and Suggestions for Authors

This is an interesting review summarizing the research conducted on the importance of chemokines in the pathophysiology of AML. A few points to be addressed by the authors:

(a) Please include a table summarizing the current classification of AML in comparison with the FAB which is a old classification but still used in clinical practice so as the average reader gets familiar with the spectrum of the disease. Some studies referred by the authors still use the old FAB classification and this add some confusion about the importance of those studies. 

(b) Please include the keywords employed in the research of literature and the time frame used in the search. 

(c)  In each table in the paper there is a confusion about which part of column 1 is associated with column 2 and 3. Please add lines to the tables so as to be more clear. Also, add a fourth column with the corresponding references. 

(d)  A figure with the receptors of chemokines and the cascade of the second messengers could also be useful so as to understand the underlying mechanisms which may be involved in AML.

(e) Are there any studies investigating the role of chemokines in the pre-AML stages (i.e. progression to AML from MDS).  This could be also interesting and could add some important information about the pathogenesis of the disease since most of AML cases in old people are MDS related. Are there any evidence of their role in the disturbance of the marrow microenvironment that accelerates or inhibits apoptosis and to which point? 

Comments on the Quality of English Language

The paper needs editing for minor typos and grammatical errors. 

Author Response

Review 2

Comments 1: This is an interesting review summarizing the research conducted on the importance of chemokines in the pathophysiology of AML. A few points to be addressed by the authors:

(a) Please include a table summarizing the current classification of AML in comparison with the FAB which is a old classification but still used in clinical practice so as the average reader gets familiar with the spectrum of the disease. Some studies referred by the authors still use the old FAB classification and this add some confusion about the importance of those studies. 

Response 1(a): The description of the AML subtype classification has been revised.

(b) Please include the keywords employed in the research of literature and the time frame used in the search. 

Response 1 (b): description of the research process for gathering materials to write the article has been added.

(c)  In each table in the paper there is a confusion about which part of column 1 is associated with column 2 and 3. Please add lines to the tables so as to be more clear. Also, add a fourth column with the corresponding references. 

Response 1(c): The tables have been modified.

(d)  A figure with the receptors of chemokines and the cascade of the second messengers could also be useful so as to understand the underlying mechanisms which may be involved in AML.

Response 1(d): The figure has been added along with a brief description.

(e) Are there any studies investigating the role of chemokines in the pre-AML stages (i.e. progression to AML from MDS).  This could be also interesting and could add some important information about the pathogenesis of the disease since most of AML cases in old people are MDS related. Are there any evidence of their role in the disturbance of the marrow microenvironment that accelerates or inhibits apoptosis and to which point? 

Response 1(d): The significance of the discussed chemokines for MDS has been added. Consequently, the title of the paper has been changed.

Round 2

Reviewer 1 Report

Comments and Suggestions for Authors

I read the authors response and the revised manuscript. No further questions

Reviewer 2 Report

Comments and Suggestions for Authors

All my comments have been succesfully addressed. I have no further comments.